# The Role of Clinical Pharmacists in Improving Quality of Care in Patients with Inflammatory Bowel Disease: An Evaluation of Patients’ and Physicians’ Satisfaction

**DOI:** 10.3390/healthcare10101818

**Published:** 2022-09-21

**Authors:** Fatema Alrashed, Najlaa Almutairi, Mohammad Shehab

**Affiliations:** 1Department of Pharmacy Practice, Faculty of Pharmacy, Health Sciences Center (HSC), Kuwait University, Jabriya 13110, Kuwait; 2Department of Internal Medicine, Mubarak Al-Kabeer Hospital, Faculty of Medicine, Kuwait University, Jabriya 47060, Kuwait

**Keywords:** IBD, clinical, pharmacist

## Abstract

Background: Inflammatory bowel disease (IBD) is a chronic and complex disease that requires a multidisciplinary team, including clinical pharmacists, to improve the quality of care and patient outcome. This present study aimed to assess the satisfaction of patients with IBD and physicians regarding clinical pharmacist interventions in outpatient and inpatient settings. Methods: A survey-based study was performed between 1 March and 1 August 2022 in a tertiary care IBD center. Two different questionnaires were distributed among patients and physicians focusing on satisfaction with the clinical pharmacist services. Patient demographics were obtained. Descriptive statistics were used to summarize the results of the survey. Results: A total of 108 patients with IBD and 23 physicians participated in this study. Among study participants, Crohn’s disease (CD) accounted for 64.8% of the total participants, while 35.2% of patients had ulcerative colitis (UC). Regarding the patient survey, most patients were extremely satisfied with clinical pharmacists’ services, during which the majority strongly agreed or agreed that they were satisfied with the counseling session. However, five patients were unsure about the amount of time spent with the clinical pharmacist. There were no patients dissatisfied with any of the services. Finally, two physicians were not sure regarding clinical pharmacists monitoring patients’ responses in of terms of toxicity and adverse effects. Conclusions: the current study illustrates patients’ and physicians’ high satisfaction with clinical pharmacists’ services in outpatient and inpatient settings. The findings of this study as well as previous studies necessitate expanding the clinical pharmacist services in the gastroenterology field.

## 1. Introduction

Inflammatory bowel diseases (IBD), which include Crohn’s disease (CD) and ulcerative colitis (UC), are multiple disorders that are characterized by the inflammation of the gastrointestinal (GI) tract. IBD is a chronic, relapsing, and remitting disease that frequently necessitates long-term therapy and regular monitoring [1]. In treating IBD, the determined goals must be achievable and realistic for each patient. Although the medications in IBD treatment are considered effective, they also have well-known, potentially severe adverse effects [2,3]. Thus, the treatment plan for each patient needs to be individualized depending on factors such as age and severity of the disease to maximize the efficacy and avoid serious complications associated with the disease and medications [2]. Because each treatment plan must be individualized to the patient, this leads to an increase in the utilization of healthcare resources, including the provider’s time [4]. Additionally, over the years, IBD has become a significant global burden, and its impact on healthcare systems will grow dramatically as the number of cases rises [5]. Furthermore, IBD is one of the top five most costly gastrointestinal diseases with high societal costs and a lower quality of life [6].

Due to the challenges associated with IBD management, a multidisciplinary team, including a clinical pharmacist, is required to achieve remission, avoid complications, improve the patient’s quality of life, and decrease the associated economic burden [7]. The American College of Clinical Pharmacy (ACCP) defines the clinical pharmacist as the direct patient care provider who works collaboratively with the medical team to optimize medication use and promote health [8]. The clinical pharmacist’s function as a part of the GI unit team is diverse. It involves managing all aspects related to medication, such as therapeutic drug monitoring (TDM), and initiating or altering pharmaceutical therapy care plans based on different patient factors and responses. Moreover, the clinical pharmacist can closely and continuously monitor the medication outcome to ensure safe and effective use. The clinical pharmacist can also provide patients with accurate information, education, and counseling regarding the medication, ensures patient adherence by addressing any concerns, and communicates any present issues with the whole team [9]. International and regional studies performed to assess patients’ satisfaction with the various services the pharmacist delivers demonstrated how the role of clinical pharmacists has changed over the years and now clinical pharmacists have become an integral part of the healthcare team [10,11,12,13]. However, in Kuwait, the concept of a clinical pharmacist is relatively new and evolving. Pharmacists’ duties in both the public and private sectors are still focused on dispensing with some clinical pharmacy activities based on individual initiatives from motivated pharmacists. This now is slowly changing as the Ministry of Health in Kuwait is starting to introduce a detailed job description for clinical pharmacists. Furthermore, clinical pharmacy units are being established in some hospitals in Kuwait, however, due to the limited number of clinical pharmacists, not all inpatient units or outpatient clinics in a hospital include clinical pharmacists as part of their multidisciplinary team, which leaves clinical pharmacists rotating between different units to provide clinical services. In addition, historically, physicians in Kuwait were comfortable with pharmacists providing a broad range of services but appear somewhat less comfortable with pharmacists’ provision of direct patient care [14]. However, a more recent study showed that this is gradually changing [15]. Physicians now consider pharmacists as integral members of the healthcare team and recognize that they can improve quality of clinical care. Nevertheless, it is difficult to gauge patients’ or physicians’ satisfaction with the growing specialized clinical pharmacy services due to lack of research. Thus, this study aims to evaluate patients’ and physicians’ satisfaction in relation to the services provided by the clinical pharmacists to patients with IBD in outpatient clinics as well as inpatient gastroenterological units.

## 2. Materials and Methods

### 2.1. Study Design and Settings

A survey-based study was performed between 1 March and 1 August 2022 in a tertiary care gastroenterology center at Mubarak Alkabeer Hospital. The study was performed at outpatient clinics and gastroenterology inpatient units. For the purpose of this study, two types of self-administered questionnaires, targeting patients with IBD and physicians at the gastroenterology inpatient units, were used. The questionnaires were validated in a previous study [12] and underwent internal review to assess for reliability and validity by performing test–retest reliability. During the internal review, the questionnaire (test) was distributed to ten patients randomly in the clinic, and, after 4 weeks, the questionnaire (retest) was given to the same patients to ascertain the reliability of the questionnaire. In addition, the questionnaires were designed in Arabic and English to evaluate patients’ and physicians’ satisfaction with the services provided by clinical pharmacists, and they were validated using the forward-backward method. Two native Arabic speakers independently performed a forward translation. This version was translated back into English independently by two native English speakers with no medical background.

Inclusion criteria for participants were patients: (1) 18 years of age or older, (2) diagnosed with IBD for at least 8 weeks since recruitment, and (3) have been counseled by a clinical pharmacist for at least one session. Inclusion criteria for physicians were gastroenterologists or internal medicine residents who have been working in the inpatient gastroenterology unit for at least 4 weeks. Informed consent was obtained by participants before distributing the questionnaires. Participants who were not willing to participate were excluded. 

Clinical pharmacists were defined as pharmacists who have a Doctor of Pharmacy (Pharm. D.) degree or have completed one or more years of postgraduate training after completing a Bachelor of Pharmacy degree (BPharm). 

Diagnosis of IBD was made according to the international classification of diseases (ICD-11 version: 2022) [16]. Patients were considered to have IBD when they had ICD-11 K50, K50.1, K50.8, and K50.9 corresponding to Crohn’s disease (CD) or ICD-11 K51, K51.0, K51.2, K51.3, K51.5, K51.8, and K51.9 corresponding to ulcerative colitis (UC). The study was performed and reported in accordance with the Strengthening the Reporting of Observational Studies in Epidemiology (STROBE) guidelines for a qualitative assessment of content and a survey of endorsement [17]. The study was approved by the local Ministry of Health and Ethical Review Board, reference: 127, protocol number 143/2022.

### 2.2. Survey Measures

The physician and patient questionnaires comprised a total of eight statements to assess their satisfaction. A 5-point Likert scale was used to evaluate their satisfaction with each statement: strongly agree, agree, not sure, disagree, and strongly disagree, and the patients or the physicians chose what best described their satisfaction with the clinical pharmacists. The patient questionnaire had a component to obtain information on the demographics of participants.

The patient questionnaires were given after the clinical pharmacist counseling session at the outpatient clinic. A pharmacy intern was responsible for distributing and collecting the questionnaires during the counseling session. Then, a verbal description of the questionnaire was performed by the pharmacy intern before distributing them. Patients were given an explanation of the purpose of the study and overview of all the statements included in the questionnaire. Moreover, patient demographic data, including date of birth, weight, height, gender, current medications, and co-morbidities, were collected by asking them verbally along with the patient questionnaire. The subjects completed the questionnaire anonymously. The patient questionnaire mainly focuses on the clinical pharmacist’s performance after the counseling session. Counseling sessions focused on the proper use of IBD medications as well storage information and any potential adverse events or drug–drug interactions. In addition, the clinical pharmacists addressed any questions or concerns patients with IBD had regarding their drug therapy. 

The physician questionnaire was only distributed to those that worked directly with the clinical pharmacist in the inpatient gastroenterology units. Similarly, the aims of the study and the questionnaire were verbally explained to the physicians. The questionnaire gathered information concerning the level of satisfaction of the physicians with the overall services performed by clinical pharmacists in the inpatient settings regarding the care of patients with IBD. The survey was distributed anonymously to the physicians with no names or other personal information identified before starting their shift or during their breaks. 

The outcomes of this study were evaluated from the data filled by patients and physicians answering the questionnaires. The responses of satisfaction-related questions were assessed by estimating the percentages of participants answering the individual responses ranging from strongly agree to strongly disagree.

### 2.3. Statistics 

We performed descriptive statistics to characterize the study. Standard descriptive statistics were used to present the demographic characteristics of patients included in this study. Descriptive statistics, including frequencies and ranges, were calculated for all survey variables. Analysis was conducted using Stata software version 15.1 (Stata Corp LP, College Station, TX, USA).

## 3. Results

### 3.1. Patients Demographics

A total of 108 patients with IBD were enrolled in the study. In addition, 20 physicians agreed to participate in the study and answered the questionnaire. Patients’ demographics are represented in Table 1. Mean age and BMI were 33 years old and 26.7 kg/m^2^. Most of the participants were male (55.5%). More than half of the patients had Crohn’s disease (CD) (64.8%), and approximately 35.2% had ulcerative colitis (UC). For the drugs that were used in the IBD treatment, most of the patients were on adalimumab (21.3%) and aminosalicylates (18.5%). One patient was receiving tofacitinib. Moreover, cardiovascular disease accounted for the majority of co-morbidities (17.5%). 

### 3.2. Outcomes

Regarding the patient surveys, most patients were extremely satisfied with clinical pharmacists’ services, during which the majority strongly agreed or agreed that they were satisfied with the counseling session. However, five patients were unsure about the amount of time spent with the clinical pharmacist. Finally, there were no patients dissatisfied with any of the services. The level of satisfaction of IBD patients is shown in Table 2.

Table 3 shows the response of the physicians to the questionnaire. Most of the providers revealed positive perceptions and satisfaction levels with the inpatient services performed by the clinical pharmacist. Nevertheless, there were two physicians who were not sure about the ability of the clinical pharmacist to monitor the patient’s response in terms of toxic and side effects of IBD drugs. Similarly, there was no disagreement with any of the services that clinical pharmacists can perform in inpatient settings. 

## 4. Discussion

The current study was carried out among IBD patients to measure their satisfaction with clinical pharmacists’ outpatient services. Patient satisfaction is an essential indicator and humanistic outcome that healthcare providers use to assess the quality of their services from the patients’ perspective. Additionally, it evaluates if the patient’s expectations and requirements for a particular service are met. Moreover, it is an important healthcare quality aspect that needs to be considered when establishing long-term healthcare services to ensure the continuance of the provided service [18]. Additionally, adapting adjustments to the services based on patients’ satisfaction will help identify the aspects that need improvement, as well as enhance beneficial modifications in the present service. This will improve the healthcare system and ensure that the service outcome addresses the patient’s requirements [19]. In addition, some studies have linked patient satisfaction with medication adherence and compliance [20].

For patients generally, the outcomes of this research demonstrate a good level of satisfaction among the respondents with the counseling session that the clinical pharmacist provided. The current study results are equivalent to some international and regional studies done to assess patients’ satisfaction with the various services the pharmacist delivers. This includes pharmacists in community pharmacy [13,19,21], as well as in the medical wards [22] and outpatient settings [23]. Overall, these findings imply that the majority of patients were pleased with these services. In comparison, some regional studies concluded that patients were unsatisfied with the performed services of the pharmacists [24,25]. The different study results may be related to factors that can influence patient satisfaction, such as a pharmacist’s qualification, training experience, and different population of each study.

In this study, the lowest degree of satisfaction was observed in the patients’ time spent with clinical pharmacists. Five patients were unsure about the time, and the others tended to agree only. This outcome is in line with what was demonstrated by Jose et al. in a study conducted in Oman [12]. Additionally, in a study performed in Saudi Arabia, among all the participants, nearly half of them only agreed that the pharmacist spent sufficient time with them in the community pharmacy [13]. This may be related to the workload of the pharmacist and the amount of time available for each patient to explain all information related to the drugs or the disease. Moreover, the ability of each patient to digest the information given by the clinical pharmacist varies significantly. Additionally, in IBD, the drug regimen tends to be more complex in terms of frequency and route of administration. For example, the frequency of some medications can change on a weekly basis, and the route of administration of some medications, such as enema or suppositories, commonly used in IBD patients’ needs to be explained thoroughly to achieve therapeutic outcomes and ensure remission. On the other hand, the highest satisfaction level was noticed regarding the language that is used by the pharmacists while discussing any issue related to drugs. Similarly, the ward pharmacists’ language was rated as satisfactory by about 96% of participants in a study done in Malaysia [22]. This is controversial to a survey that was undertaken in Oman, during which some individuals were dissatisfied with the pharmacist’s language while counseling. The authors suggest that a possible cause is that the majority of pharmacists were non-Arabic speakers in the Oman study [12], while, in both Malaysia and this study, the clinical and ward pharmacists could communicate in several languages, including their mother language. This is important because if the patients clearly understand the instructions and the clinical pharmacist is aware of their concerns and answers their questions, this can ultimately lead to better outcomes in terms of adherence and satisfaction, leading to better results. 

In relation to patient demographics, despite the inconsistent results in some studies, patient satisfaction has been connected to demographic factors and other factors such as waiting time, the environment, and the quality of provided services [10,26]. Furthermore, in this study, the patients expressed high satisfaction regarding privacy during the counseling session and the clinical pharmacists’ questions regarding medications. For privacy, comparing these results to earlier regional studies, there was a significant change in satisfaction since the clinical pharmacists performed their counseling in a private room without any distractions [24,25]. This proves that privacy is an important point that needs to be considered in a counseling session with any patient, especially in IBD, since counseling is essential for achieving the best possible results.

Our study also investigated physicians’ satisfaction with clinical pharmacists’ services in gastroenterology inpatient units. According to the results of the questionnaire, physicians believe that involving clinical pharmacists in the healthcare team has a beneficial influence. Moreover, the results of our study showed great degrees of satisfaction with clinical pharmacists’ services, including inpatient intervention in the gastroenterology inpatient unit. The majority of physicians that participated in the study strongly agreed or agreed, as shown in Table 3. Other studies have found that primary care physicians expressed high levels of satisfaction with the clinical pharmacists helping in their practice [7,27]. All the responders in our study believed that working collaboratively with clinical pharmacists could improve the overall quality of care and patient outcome. Similar results were found in several regional studies [28,29]. Additionally, the findings of the physicians’ survey revealed that clinical pharmacists have an essential role in counseling and patient education regarding their pharmacotherapy. This is also in agreement with findings from other studies [28,30]. When it comes to presenting drug information to healthcare professionals, the findings are consistent with previous research [14,31]. This agreement was noticed in almost all of the studies as part of the traditional role that pharmacists are always known to be the experts in any medication aspect.

The physicians also showed acceptance of clinical pharmacist interventions and recommendations in drug therapy management. The results of this study are also comparable with the other two studies that were conducted in Kuwait; one was conducted in a private hospital and the other in different public hospitals. They found that the treating physicians desired clinical pharmacists’ interventions for drug-related issues [15,32]. However, in a recent study that was conducted in Saudi Arabia, the authors found that some consultant physicians were less comfortable with clinical pharmacists’ intervention regarding any adjustment that is done by the physicians. The authors suggest that this may be related to their belief that clinical pharmacists are overreaching their role and impede their decisions. In addition, they noticed that newly graduated physicians in practice were more likely to accept the recommendation from the clinical pharmacists [29]. Another possible cause is that the clinical pharmacist role is still in development, and it is not completely clear to some physicians.

Another study illustrated that physicians’ perceptions of clinical pharmacists’ roles have improved in recent years, and their expectations toward their services extend beyond counseling and dispensing. In addition, they recognized clinical pharmacists as an integral part of the medical team to achieve the desired therapeutic outcome [15]. Comparing a study that was undertaken in 2004 in Kuwait, most physicians do not expect the clinical pharmacists to aid in developing drug therapy treatment plans and interfere with patient monitoring to assess the efficacy and safety of the medications. Additionally, only almost half of them expected the pharmacists to be available during ward round [14]. This change in appreciation of the clinical pharmacist’s role may be related to the fact that, in this study, physicians worked directly with the clinical pharmacists, which may have changed their perspective and confidence.

In recent years, pharmacists’ roles and responsibilities have evolved to focus more on direct patient care. In Kuwait, the expanded part of pharmacy practice is transforming slowly, and interdisciplinary management teams are not frequently used in all the hospitals. Involving clinical pharmacists in managing patients with IBD shows a significant decrease in drug errors and improved quality of care [4]. In addition, prospective, interventional follow-up research performed in an IBD outpatient clinic to assess clinical pharmacists’ interventions on patients’ compliance and their awareness regarding medications enrolled about 110 patients. The authors found that enhancing patient medication understanding through clinical pharmacists’ interventions led to better drug adherence [33]. In addition, the clinical pharmacist’s positive impact can extend to include even other healthcare providers, including physicians. For example, from a physician’s point of view, the presence of clinical pharmacists in the care team can relieve some of their burdens, allowing them to focus on other aspects of the disease and ensure comprehensive medication management [34]. This evidence along with our study emphasize the importance of having a clinical pharmacist involved in caring for patients with IBD. 

Because there is limited regional research regarding clinical pharmacists’ role in caring for patients with IBD, our study is considered the first study in Kuwait to assess physicians’ satisfaction with clinical pharmacists’ services in inpatient and outpatient settings. Additionally, results of the present study allow future researchers to build on the findings. However, there are limitations to this research. First, the findings of this study cannot be generalized to all the patients and physicians in Kuwait because it was conducted in a single hospital. Furthermore, the study duration was short; this did not allow for surveying more patients and having a larger sample size. 

This study specifically evaluated the satisfaction of physicians and patients, future studies can perhaps explore the perception of IBD clinical pharmacists to better understand the services they can provide to this group of patients. Finally, a larger multicentered study exploring whether the IBD clinical pharmacists influenced patient’s specific outcomes, such as improving adherence, meeting treatment goals, and enhancing patients’ quality of life, will be helpful in providing a clear picture of the impact of IBD clinical pharmacists. 

## 5. Conclusions

In conclusion, the majority of participants demonstrated a high level of satisfaction with clinical pharmacists’ involvement and services in IBD clinics and gastroenterology inpatient units. However, more studies with a larger population and in different hospitals are needed to have better understanding about the perceptions of both patients and physicians. Moreover, the role of clinical pharmacists in caring for patients with IBD needs to be established in order to improve the quality of care of this group of patients.

## Figures and Tables

**Table 1 healthcare-10-01818-t001:** Demographic characteristics of the patients.

The Variable	Total *n*= 108
Age	
15–35 years old	67 (62.0%)
36–55 years old	35 (32.4%)
>55 years old	6 (5.6%)
Gender *n* (%)	
Male	60 (55.5%)
Female	48 (44.5%)
BMI	
Mean BMI kg/m^2^ (SD)	26.7 (±5)
Co-morbidities *n* (%)	
Asthma	9 (8.3%)
Chronic Obstructive Pulmonary Disease	12 (11.1%)
Cardiovascular Disease	19 (17.5%)
Diabetes	15 (13.8%)
Hypertension	12 (11.1%)
Others	22 (20.3%)
None	19 (17.8%)
Type of IBD *n* (%)	
CD	70 (64.8%)
UC	38 (35.2%)
Current medications for IBD * *n* (%)	
Infliximab	15 (13.8%)
Ustekinumab	17 (15.7%)
Azathioprine	18 (16.6%)
Aminosalicylates	20 (18.5%)
Budesonide	5 (4.6%)
Adalimumab	23 (21.3%)
Vedolizumab	10 (9.2%)
Methotrexate	1 (0.9%)
Tofacitinib	1 (0.9%)

* Some patients were on more than one medication for IBD.

**Table 2 healthcare-10-01818-t002:** Patients’ responses regarding their level of satisfaction.

Disagree	Strongly Disagree	Not Sure	Agree	Strongly Agree	Question
0	0	0	19 (17.4%)	89 (82.6%)	I am satisfied with the type and amount of information discussed by the pharmacist on drugs related to inflammatory bowel disease
0	0	0	33 (30.5%)	75 (69.5%)	I am satisfied with the questions asked by my pharmacist regarding my medications like any history of previous drug allergy, disease details, etc
0	0	0	19 (17.4%)	89 (82.6%)	I am satisfied with the privacy maintained by pharmacist while discussing with patients and dispensing medications
0	0	0	19 (17.4%)	89 (82.6%)	I am satisfied with the level of knowledge that pharmacists demonstrate in drugs related to inflammatory bowel disease
0	0	0	19 (17.4%)	89 (82.6%)	I am satisfied with the kind of response pharmacist provide on questions related to drugs I am taking for inflammatory bowel disease
0	0	0	14 (13%)	94 (87%)	I am satisfied with the language used by the pharmacist in discussing drug related matters
0	0	5 (4%)	28 (26%)	75 (70%)	I am satisfied by the amount of time spend by my pharmacist with each patient
0	0	0	14 (13%)	94 (78.3%)	I am satisfied with the kind of information the pharmacist provides on inflammatory bowel disease and other health issues along with information on drugs

**Table 3 healthcare-10-01818-t003:** Physicians’ responses regarding their level of satisfaction.

Disagree	Strongly Disagree	Not Sure	Agree	Strongly Agree	Question
0	0	0	4 (20.0%)	16 (80.0%)	Clinical pharmacists’ participation in medical ward round is desirable
0	0	0	4 (20.0%)	16 (80.0%)	Clinical pharmacists can play important role in patient education and counselling
0	0	2 (10.0%)	3 (15.0%)	15 (75.0%)	Clinical pharmacists can monitor patient response to drug therapy from toxicity/side effects perspective
0	0	0	4 (20.0%)	16 (80.0%)	Clinical pharmacists can monitor patient response to drug therapy from effectiveness perspective
0	0	0	11 (55.0%)	9 (45.0%)	Clinical pharmacists can provide drug information to health care professionals such as compatibility, stability, storage, availability
0	0	0	5 (25.0%)	15 (75.0%)	Clinical pharmacy service enhances patients’ appreciation and satisfaction
0	0	0	5 (25.0%)	15 (75.0%)	Clinical pharmacists analyze patient treatment and suggest changes of therapy when necessary
0	0	0	4 (20.0%)	16 (80.0%)	Clinical pharmacists improve overcall patient outcome/quality of patient care

## Data Availability

The data presented in this study are available on request from the corresponding author. The data are not publicly available due to restrictions (privacy of patient data).

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
