# Peer review of "The Role of Clinical Pharmacists in Improving Quality of Care in Patients with Inflammatory Bowel Disease: An Evaluation of Patients’ and Physicians’ Satisfaction"

_healthcare, 2022, doi:10.3390/healthcare10101818_

Round 1
Reviewer 1 Report
The manuscript tackles an interesting topic of the role of the pharmacists in the treatment of the IBD patients, as a part of a multidisciplinary team. The main goal was to assess the satisfaction of the patients and physicians with pharmacists’ services. The study was designed as an observational, cross-sectional study.
Introduction:
Introduction was well written, covered the main background points and led up to the aim of the study.
Methods:
Authors mentioned that they conducted the internal review to assess the questionnaires’ validity and reliability. Please report the results of this review, preferably as test-retest reliability and internal reliability.
Authors reported they used previously developed questionnaire from the study Jose J, et al.: Public’s perception and satisfaction on the roles and services provided by pharmacists 351 - Cross sectional survey in Sultanate of Oman. Saudi Pharmaceutical Journal. 2015;23(6):635-641. In original questionnaire by Jose et al., they used 9 items to test patient satisfaction while the authors in this study used 8 items, omitting the item “I am satisfied with the relationship that the pharmacist tries to maintain with the patients”. Please elaborate why was this item removed from your questionnaire as it seems relevant to this topic.
Also, in the original study they investigated the perception of the pharmacists by the patients, which would also be relevant and interesting for this study, to see how patients and physicians perceive pharmacists in Kuwait, but was unfortunately not conducted in this study.
Authors only conducted the satisfaction questionnaire, and did not report if the patients met their treatment goals, changes in their quality of life, drug adverse events or any other outcome measures to see if the results of the satisfaction questionnaire were correlated with treatment success which would make this study more informative and relevant.
Study is single centre only, with a limited sample size, which was addressed in the limitation section of the Discussion. Still, the sample size of the physicians is too small to be of any relevance. Sample size calculation was not conducted.
Page 3 lines 103-104. Please elaborate the sentence: “Then, a verbal discerption of the questionnaire is done before disturbing them.”
Results and discussion:
Results section reported all the data described in the methods section and the results were mostly adequately interpreted and given the broader context in the discussion section.
Page 6 lines 194-195 By “as in the medical words” you probably meant medical wards?
This is an interesting study that could be of interest to the readers. Unfortunately, it has certain flaws that need to be addressed by the authors.
Author Response
Reviewer 1
Comments and Suggestions for Authors
The manuscript tackles an interesting topic of the role of the pharmacists in the treatment of the IBD patients, as a part of a multidisciplinary team. The main goal was to assess the satisfaction of the patients and physicians with pharmacists’ services. The study was designed as an observational, cross-sectional study.
Introduction:
Introduction was well written, covered the main background points and led up to the aim of the study.
Methods:
Authors mentioned that they conducted the internal review to assess the questionnaires’ validity and reliability. Please report the results of this review, preferably as test-retest reliability and internal reliability.
Thank you for your comment. We have added details of the internal review in the method section.
Method:
“The questionnaires were validated in a previous study and underwent internal review to assess for reliability and validity by performing test-retest reliability. During the internal review, the questionnaire (test) was distributed to ten patients randomly in the clinic, after 4-weeks-point, the questionnaire (retest) was given to the same patients to ascertain the reliability of the questionnaire.”
Authors reported they used previously developed questionnaire from the study Jose J, et al.: Public’s perception and satisfaction on the roles and services provided by pharmacists 351 - Cross sectional survey in Sultanate of Oman. Saudi Pharmaceutical Journal. 2015;23(6):635-641. In original questionnaire by Jose et al., they used 9 items to test patient satisfaction while the authors in this study used 8 items, omitting the item “I am satisfied with the relationship that the pharmacist tries to maintain with the patients”. Please elaborate why was this item removed from your questionnaire as it seems relevant to this topic.
Thank you for our comment. We have intentionally removed this item from the questionnaire because the clinical pharmacists at the gastroenterology unit/clinic are rotating on 6-weeks basis, therefore, we felt that this specific item does not apply to our cohort.
Also, in the original study they investigated the perception of the pharmacists by the patients, which would also be relevant and interesting for this study, to see how patients and physicians perceive pharmacists in Kuwait, but was unfortunately not conducted in this study.
Thank you for our comment. Indeed, the perception of pharmacists by patients is very interesting as well as important. However, the aim of our study was very specific to the satisfaction of patients and physicians. We did add this point to the discussion section.
“This study specifically evaluated the satisfaction of physicians and patients, future studies can perhaps explore the perception of IBD clinical pharmacists to better understand the services they can provide to this group of patients. Finally, a larger multicentered study exploring whether the IBD clinical pharmacists influenced patient’s specific outcomes such as improving adherence, meeting treatment goals, and enhancing patients’ quality of life will be helpful in providing a clear picture of the impact of IBD clinical pharmacists.”
Authors only conducted the satisfaction questionnaire, and did not report if the patients met their treatment goals, changes in their quality of life, drug adverse events or any other outcome measures to see if the results of the satisfaction questionnaire were correlated with treatment success which would make this study more informative and relevant.
Thank you for our comment. We have added this to the discussion section.
“This study specifically evaluated the satisfaction of physicians and patients, future studies can perhaps explore the perception of IBD clinical pharmacists to better understand the services they can provide to this group of patients. Finally, a larger multicentered study exploring whether the IBD clinical pharmacists influenced patient’s specific outcomes such as improving adherence, meeting treatment goals, and enhancing patients’ quality of life will be helpful in providing a clear picture of the impact of IBD clinical pharmacists.”
Study is single centre only, with a limited sample size, which was addressed in the limitation section of the Discussion. Still, the sample size of the physicians is too small to be of any relevance. Sample size calculation was not conducted.
Thank you for your comment. In this study we aimed to assess the satisfaction of patients and physicians using descriptive statistics, including frequencies and ranges, we did not attempt to calculate any associations. In addition, previous studies sample size varied from 42 patients (Moreno et al) to 390 patients (Jose et al) with no sample size calculations.
- Jose J, al Shukili MN, Jimmy B. Public’s perception and satisfaction on the roles and services provided by pharmacists - Cross sectional survey in Sultanate of Oman. Saudi Pharmaceutical Journal. 2015;23(6):635-641. doi:10.1016/j.jsps.2015.02.003
- Moreno G, Lonowski S, Fu J, et al. Physician experiences with clinical pharmacists in primary care teams. Journal of the American Pharmacists Association. 2017;57(6):686-691. doi:10.1016/j.japh.2017.06.018
Page 3 lines 103-104. Please elaborate the sentence: “Then, a verbal discerption of the questionnaire is done before disturbing them.”
Thank you for your comment, we have added the details to the method section.
“A pharmacy intern was responsible for distributing and collecting the questionnaires. Then, a verbal description of the questionnaire was performed by the pharmacy intern before distributing them. Patients were given an explanation of the purpose of the study and overview of all the statements included in the questionnaire.”
Results and discussion:
Results section reported all the data described in the methods section and the results were mostly adequately interpreted and given the broader context in the discussion section.
Page 6 lines 194-195 By “as in the medical words” you probably meant medical wards?
Thank you for your comment, yes we have now corrected it.
This is an interesting study that could be of interest to the readers. Unfortunately, it has certain flaws that need to be addressed by the authors.
Thank you, we appreciate the time taken to review this paper.
Reviewer 2 Report
General comments:
Thank you for the opportunity to review this manuscript.
Whilst the authors attempted to evaluate the patients’ and physicians’ satisfaction in relation with the services provided by the clinical pharmacist in the IBD clinic, the methodology employed lacks rigor and does not specifically address the aim.
The authors could consider investigating patients’ medication adherence to identify if there is any correlation between patient satisfaction and medication adherence. Alternatively, a follow-up study to evaluate attitudes and perceptions of patients and physicians towards the role of IBD pharmacists using a mixed methods study design e.g. questionnaires and interview/focus group could be considered to increase the merits of this work.
Abstract:
The percentages of patients with CD and UC do not add up to 100%.
Introduction:
The introduction lacks information on the role of IBS pharmacist at the Mubarak Alkabeer Hospital. Although the authors stated that the concept of clinical pharmacist is relatively new in Kuwait, an article published in 2006 on “Physicians’ Perceptions and Expectations of Pharmacists’ Professional Duties in Government Hospitals in Kuwait” by Matowe L. et al. (DOI: 10.1159/000092179), has evaluated physicians’ perceptions, expectations and experiences with pharmacists. Is the role of an IBD pharmacist a recent addition to this hospital or in Kuwait? More details are required to provide readers context on why this study is important.
Methods:
Was there any validation of the questionnaires in Arabic and English to ensure consistency?
What was entailed in the counselling session in the outpatient clinic? Was it specific to IBD counselling or a general counselling session offered to all outpatient patients? Were the counselling sessions carried out by a single pharmacist or multiple pharmacists?
It was unclear if the IBD clinical pharmacist or another staff member administered the patient questionnaires after the clinical pharmacist counselling session. If the clinical pharmacist who delivered the counselling session gave the questionnaires, the participants may be compelled to complete the questionnaire or provide good feedback, even though it is anonymous. This could potentially skew the results significantly.
This sentence is unclear “….verbal discerption of the questionnaire…..”
It appears that there is only one clinical pharmacist working in the inpatient gastroenterology unit. The physicians may also be compelled to provide good back for the service provided by the pharmacist.
Is there a reason why the questionnaire focused on the level of satisfaction of the physicians working at the gastroenterology unit with overall services carried out by clinical pharmacists, rather than on IBD-related services? This contradicts with the aim stated in the introduction – which is to evaluate patients’ and physicians’ satisfaction in relation with the services provided by the clinical pharmacist in the IBD clinic.
Results:
Age of participants should be presented as different age groups, rather than as a mean.
Figure 1 is duplicating data in Table 1 and is unnecessary.
There is a discrepancy in the numbers presented for question 7 between Table 1 and Figure 1. Is it supposed to be 75,28,5,0,0 or 75,28,1,0,0?
Similarly, Figure 2 is duplicating data in Table 2 and should be removed.
Discussion:
It should be Malaysia not Malesia.
Correlation study between demographic characteristics and patients’ satisfaction should be included in the analysis of this study, to strengthen the findings of this study.
Mechanics & Referencing:
The text requires a thorough check to ensure that it is free from grammar/spelling errors.
There is also an error in referencing - Reference 2 is incomplete.
Author Response
Reviewer 2
Thank you for the opportunity to review this manuscript.
Whilst the authors attempted to evaluate the patients’ and physicians’ satisfaction in relation with the services provided by the clinical pharmacist in the IBD clinic, the methodology employed lacks rigor and does not specifically address the aim.
The authors could consider investigating patients’ medication adherence to identify if there is any correlation between patient satisfaction and medication adherence. Alternatively, a follow-up study to evaluate attitudes and perceptions of patients and physicians towards the role of IBD pharmacists using a mixed methods study design e.g. questionnaires and interview/focus group could be considered to increase the merits of this work.
Thank you for our comment. We have added this to the discussion section.
“This study specifically evaluated the satisfaction of physicians and patients, future studies can perhaps explore the perception of IBD clinical pharmacists to better understand the services they can provide to this group of patients. Finally, a larger multicentered study exploring whether the IBD clinical pharmacists influenced patient’s specific outcomes such as improving adherence, meeting treatment goals, and enhancing patients’ quality of life will be helpful in providing a clear picture of the impact of IBD clinical pharmacists.”
Abstract:
The percentages of patients with CD and UC do not add up to 100%.
Thank you for your comment, we have corrected it.
“Among study participants, Crohn’s disease (CD) accounted for 64.8% of the total participants, while 35.2% of patients had ulcerative colitis (UC).”
Introduction:
The introduction lacks information on the role of IBS pharmacist at the Mubarak Alkabeer Hospital. Although the authors stated that the concept of clinical pharmacist is relatively new in Kuwait, an article published in 2006 on “Physicians’ Perceptions and Expectations of Pharmacists’ Professional Duties in Government Hospitals in Kuwait” by Matowe L. et al. (DOI: 10.1159/000092179), has evaluated physicians’ perceptions, expectations and experiences with pharmacists. Is the role of an IBD pharmacist a recent addition to this hospital or in Kuwait? More details are required to provide readers context on why this study is important.
Thank you for your comment. We have added it to the introduction section.
“Pharmacists’ duties in both the public and private sectors are still focused on dispensing with some clinical pharmacy activities based on individual initiatives from motivated pharmacists. This now is slowly changing as the Ministry of Health in Kuwait is starting to introduce a detailed job description for clinical pharmacists. Furthermore, clinical pharmacy units are being established in some hospitals in Kuwait, however, due to the limited number of clinical pharmacists, not all inpatients’ units or outpatients’ clinics in a hospital include clinical pharmacists as part of their multidisciplinary team, which leaves clinical pharmacists rotating between different units to provide clinical services. In addition, historically, physicians in Kuwait were comfortable with pharmacists providing a broad range of services but appear somewhat less comfortable with pharmacists’ provision of direct patient care.14 However, a more recent study showed that this is gradually changing.15 Physicians now consider pharmacists as integral members of the healthcare team and recognize that they can improve quality in patient clinical care. Nevertheless, it is difficult to gauge patients’ or physicians’ satisfaction with the growing specialised clinical pharmacy services due to lack of research. Thus, this study aims to evaluate patients' and physicians' satisfaction in relation with the services provided by the clinical pharmacist to patients with IBD in outpatient clinics as well as inpatient gastroenterological units.”
Methods:
Was there any validation of the questionnaires in Arabic and English to ensure consistency?
Thank you for your comment. Yes we have used the forward-backward method and we have added it to the method section.
“In addition, the questionnaires were designed in Arabic and English to evaluate patients’ and physicians’ satisfaction with the services provided by clinical pharmacists, and it was validated using the forward-backward method. Two native Arabic speakers independently performed a forward translation. This version was translated back into English independently by two native English speakers with no medical background.”
What was entailed in the counselling session in the outpatient clinic? Was it specific to IBD counselling or a general counselling session offered to all outpatient patients? Were the counselling sessions carried out by a single pharmacist or multiple pharmacists?
Thank you for your comment, the counselling session was specific to IBD medications. . And over the period of the study it was performed by different clinical pharmacists. We have now clarified this in the method section.
“Counselling sessions focused on the proper use of IBD medications as well storage information and any potential adverse events or drug-drug interactions. In addition, the clinical pharmacists addressed any questions or concerns patients with IBD had regarding their drug therapy.”
It was unclear if the IBD clinical pharmacist or another staff member administered the patient questionnaires after the clinical pharmacist counselling session. If the clinical pharmacist who delivered the counselling session gave the questionnaires, the participants may be compelled to complete the questionnaire or provide good feedback, even though it is anonymous. This could potentially skew the results significantly.
Thank you for your comment, we have added the details to the method section.
“A pharmacy intern was responsible for distributing and collecting the questionnaires during the counselling session. Then, a verbal description of the questionnaire was performed by the pharmacy intern before distributing them. Patients were given an explanation of the purpose of the study and overview of all the statements included in the questionnaire.”
This sentence is unclear “….verbal discerption of the questionnaire…..”
Thank you for your comment, we have added the details to the method section.
“A pharmacy intern was responsible for distributing and collecting the questionnaires during the counselling session. Then, a verbal description of the questionnaire was performed by the pharmacy intern before distributing them. Patients were given an explanation of the purpose of the study and overview of all the statements included in the questionnaire.”
It appears that there is only one clinical pharmacist working in the inpatient gastroenterology unit. The physicians may also be compelled to provide good back for the service provided by the pharmacist.
Thank you for your comment, different clinical pharmacists rotate on a 6-weeks basis. There is a limited number of clinical pharmacists in the hospital therefore, there is no permanent clinical pharmacist at the gastroenterology unit. In addition, the inpatient gastroenterology unit encompasses different physicians at different levels ranging from rotating residents to gastroenterologists (consultants).
Is there a reason why the questionnaire focused on the level of satisfaction of the physicians working at the gastroenterology unit with overall services carried out by clinical pharmacists, rather than on IBD-related services? This contradicts with the aim stated in the introduction – which is to evaluate patients’ and physicians’ satisfaction in relation with the services provided by the clinical pharmacist in the IBD clinic.
Thank you for your comment. We have made the objective more clear and we have also explained that the questionnaire evaluated the overall services done by clinical pharmacists in the inpatient settings regarding the care of patients with IBD only.
“Thus, this study aims to evaluate patients' and physicians' satisfaction in relation with the services provided by the clinical pharmacist to patients with IBD in outpatient clinics as well as inpatient gastroenterological units.”
“The physician questionnaire was only distributed to those that worked directly with the clinical pharmacist in the inpatient gastroenterology units. Similarly, the aims of the study and the questionnaire were verbally explained to the physicians. The questionnaire gathered information concerning the level of satisfaction of the physicians with the overall services done by clinical pharmacists in the inpatient settings regarding the care of patients with IBD”
Results:
Age of participants should be presented as different age groups, rather than as a mean.
Thank you for your comment, we have added it.
Figure 1 is duplicating data in Table 1 and is unnecessary.
Thank you for your comment. We have removed it
There is a discrepancy in the numbers presented for question 7 between Table 1 and Figure 1. Is it supposed to be 75,28,5,0,0 or 75,28,1,0,0?
Thank you for your comment, we have corrected it (75,28,5,0,0).
Similarly, Figure 2 is duplicating data in Table 2 and should be removed.
Thank you for your comment. We have removed it
Discussion:
It should be Malaysia not Malesia.
Thank you for your comment, we have corrected it.
Correlation study between demographic characteristics and patients’ satisfaction should be included in the analysis of this study, to strengthen the findings of this study.
Thank you for your comment. In this study we aimed to assess the satisfaction of patients and physicians using descriptive statistics, the power of the study was not enough to calculate any correlations between demographics characteristics and patient’s satisfaction.
Mechanics & Referencing:
The text requires a thorough check to ensure that it is free from grammar/spelling errors.
Thank you for your comment, the article has now gone through a professional english writing check.
There is also an error in referencing - Reference 2 is incomplete.
Thank you for your comment, we have completed the missing information.

Round 2
Reviewer 1 Report
The authors have performed sufficient manuscript improvements and provided adequate justification of their methods. I believe the manuscript is suitable for publication.
Author Response
Thank you for your time taken to review this article
Reviewer 2 Report
Thank you for addressing the comments. Further comments are as below.
Results:
The text does not describe the numbers in the tables correctly.
“However, one patient was unsure about the amount of time spent with the clinical pharmacist.”
There were five patients and not one patient stated in table 2.
“Nevertheless, there was one physician who was not 177 sure about the ability of the clinical pharmacist to monitor the patient’s response in terms 178 of toxic and side effects of IBD drugs.’
There were two physicians and not one physician listed in table 3.
“Methotrexate, and tofacitinib were the least prescribed medications.” Remove ,
Discussion:
Line 187 “… the patient perspective” should be patients’ perspective.
Line 207, change slightest to lowest
Line 208 -209 “One patient was unsure about the time, and the others 208 tended to agree only.” This does not match the results in Table 2.
Line 229-230 “answers their questions. This can ultimately” Replace full stop with comma, and change to this (lower case)
Line 264 – “authors” is more appropriate than “writers”
Line 267 – “overreaching” is more appropriate than “exceeding their role”; “undermine” is more suitable than “impede”
Line 274 remove “the”
Line 279 – change round to ward rounds
Line 283 the direct patient care. Remove “the”
Line 297-290. It was unclear at the start of the sentence that another study is being discussed. Please revise these sentences.
Line 300 remove “the”
Line 302 change to generalized
Line 304 – change limited to short
Author Response
Thank you for the time taken to review this article, reply to comments are attached
